# A Performance Evaluation of Convolutional Neural Network Architectures for Pterygium Detection in Anterior Segment Eye Images

**DOI:** 10.3390/diagnostics14182026

**Published:** 2024-09-13

**Authors:** Maria Isabel Moreno-Lozano, Edward Jordy Ticlavilca-Inche, Pedro Castañeda, Sandra Wong-Durand, David Mauricio, Alejandra Oñate-Andino

**Affiliations:** 1Information Systems Engineering Faculty, Universidad Peruana de Ciencias Aplicadas, Lima 15023, Peru; u201924630@upc.edu.pe (M.I.M.-L.); pcsiswon@upc.edu.pe (S.W.-D.); 2Software Engineering Faculty, Universidad Peruana de Ciencias Aplicadas, Lima 15023, Peru; u201923961@upc.edu.pe; 3Systems Engineering and Informatic Faculty, Universidad Nacional Mayor de San Marcos (UNMSM), Lima 15081, Peru; dmauricios@unmsm.edu.pe; 4Informatic and Electronics Faculty, Escuela Superior Politécnica de Chimborazo (ESPOCH), Riobamba 060155, Ecuador; monate@espoch.edu.ec

**Keywords:** pterygium detection, deep learning, Se-ResNext50, ResNext50, ResNet101, ResNext101, MobileNetV2

## Abstract

In this article, various convolutional neural network (CNN) architectures for the detection of pterygium in the anterior segment of the eye are explored and compared. Five CNN architectures (ResNet101, ResNext101, Se-ResNext50, ResNext50, and MobileNet V2) are evaluated with the objective of identifying one that surpasses the precision and diagnostic efficacy of the current existing solutions. The results show that the Se-ResNext50 architecture offers the best overall performance in terms of precision, recall, and accuracy, with values of 93%, 92%, and 92%, respectively, for these metrics. These results demonstrate its potential to enhance diagnostic tools in ophthalmology.

## 1. Introduction

Pterygium is the abnormal growth of tissue on the conjunctiva of the eye, which is the transparent membrane that covers the white part of the eye and the inner part of the eyelid. This tissue can extend into the cornea, which is the front and transparent part of the eye, which can cause discomfort when opening and closing the eye and blurred vision and even increase the risk of developing blindness or other chronic diseases [1]. While conventional methods for the detection of pterygium are effective, they are time-consuming and costly, requiring specialized medical tools and an experienced ophthalmologist to manually classify affected areas in eye images [2]. This manual procedure can affect the accuracy of the diagnosis, as the classification and interpretation of the results can be subjective, depending on the specialist’s prior experience [3].

Currently, artificial intelligence (AI) has gained relevance in many areas of health, especially in ophthalmology, due to various studies focused on detecting eye diseases such as pterygium using convolutional neural networks (CNN). In [4], a ResNet101-based model (RFRC) and a SE-ResNeXt50-based model (U-Net) were used for the detection and segmentation of pterygium, while, in [5], the VGG16 model was employed for the automatic detection of the same disease using deep learning algorithms. However, despite the promising results in eye disease detection, so far, only in [6] has the ResNext50 model been used in pterygium detection. Furthermore, in the latter, the implementation of this latest model in mobile applications has been demonstrated.

Despite the advancements achieved in pterygium detection using AI models, there are still limitations that have not been fully addressed. In particular, previous studies, such as the one presented in [6], focused on the development of a mobile application for pterygium detection using the ResNeXt50 architecture. However, this work did not consider a direct comparison with more recent models that hold the potential for superior performance, but rather limited its evaluation to the architectures most commonly used by other authors. This gap is critical, as the effectiveness and accuracy of mobile applications in clinical settings directly depend on the robustness of the underlying AI model. In this context, our study is distinguished by evaluating and comparing the performance of more advanced CNN architectures, such as ResNet101, ResNeXt101, and Se-ResNeXt50, which have demonstrated superior results in other computer vision applications. This approach will allow us to identify which of these architectures offers the best performance and, therefore, has the greatest potential for integration into the previously developed mobile application, thereby enhancing its ability to detect pterygium efficiently, accurately, and in real time. Using the same dataset obtained from the Peruvian Pterygium Center [7] and external authors [8,9,10,11], among the five evaluated models (ResNet101, ResNeXt101, Se-ResNeXt50, ResNeXt50, and MobileNet V2), the Se-ResNeXt50 architecture achieved performance values of up to 97%, 100%, 94%, and 92% in the F1 score, precision, recall, and accuracy, respectively.

This article consists of five main sections: the first presents introductory concepts within the topic; the second presents a literature review through related works; and the third section describes the models used, the creation of the dataset, the experimental settings, and the results obtained. Finally, a discussion and conclusions are provided in Section 4 and Section 5, respectively.

## 2. Related Work

Previous studies have conducted research focused on ocular diseases, including pterygium. From these studies, models with good metrics have been identified and frequently used. In [6], these models were classified into segmentation and classification. In segmentation, a model identifies and highlights the pixels in the anterior segment images of the eye, also called ASPIs, that correspond only to the area affected by pterygium. This group includes classic segmentation architectures such as fully convolutional networks (FCNs), PSPNet, SegNet, ENet, U-Net, and DenseNet, whose efficacy is guaranteed by their good metrics, as shown in [1,12]. For example, [1] reports precision of 93.33%, while [12] achieves 91.67%. Models based on architectures that are commonly used for detection have also been employed, such as in [13], where AlexNet, ResNet, and VGG16 were compared, with the latter achieving 99% precision. Another study [2] achieved precision of 91.27% using SVM for pterygium segmentation.

In classification, a model determines whether an ASPI segment of the eye belongs to a specific group or class, which commonly reflects the severity of the disease. There are numerous applications for the detection of various ocular diseases, such as diabetic retinopathy [14], glaucoma [15], and macular degeneration [16]. The most common models include VGG16 [5,8,17,18]; ResNet (and its variants) [8,17,18]; AlexNet and GoogleNet [8,17]; DenseNet201 [8]; the EfficientNet family [19]; and the MobileNet family. The architectures that stand out the most are VGG16, with 100% specificity for 772 ASPIs; EfficientNet-B6, with 99.64% specificity for 1220 ASPIs; and MobileNet2, with 98.43% specificity for 436 ASPIs.

On the other hand, ResNext50, a variant of ResNet50, has shown promising results in areas outside the ocular field, as in [20,21,22], achieving better results than other well-known models. In [20], it achieved kidney + tumor DSC of 96.78% and tumor DSC of 74.32%. Similarly, in [22], a combination of ResNext101, XceptionNet41, and DenseNet169 achieved an 87.74% F1 score and 95.75% precision in alopecia detection. In [21], they achieved 99.32% precision, 99.01% specificity, and 100% recall in the detection of COVID-19, surpassing Inception ResNet V2, ResNet50, AlexNet, Se-ResNet50, GoogleNet, and DenseNet121. In the ophthalmological field, this model was applied to diabetic retinopathy in [23], surpassing GoogleNet, AlexNet, Inception V4, and Inception ResNet V2, with 97.53% precision, 97.92% specificity, and 95.35% recall on 5333 fundus images. Likewise, in [4], a modification of this model was used for pterygium segmentation, where Liu et al. proposed a U-Net based on SE-ResNeXt50 called SRU-Net, which achieved an F1 score of 89.81%, sensitivity of 87.09%, specificity of 96.68%, and an AUC of 92.95% in slit lamp pterygium segmentation using 20,987 images and 1094 smartphone images. Finally, in [6], a mobile solution for the classification of pterygium was developed using a ResNext50 model.

From the literature review, ResNext50 has shown great application potential both within and outside the ophthalmological field. Therefore, this research carries out a comparative study of this model with others, such as ResNet101, ResNext101, Se-ResNext50, and MobileNet V2.

## 3. Materials and Methods

### 3.1. Methodology Used

Knowledge Discovery in Databases (KDD), commonly used in data mining and artificial intelligence, is a methodology that identifies useful patterns and insights from large datasets, as illustrated in Figure 1. Moreover, it consists of five stages, which were used for the implementation of the models illustrated in Figure 2. In the first stage, the data are selected; in this case, the images are referred to as ASPIs. In the second stage, these pre-selected images are manipulated to handle missing values, classify the data by class, clean the data, etc. In the third stage, the data are transformed using a data augmentation approach, seeking to generate varied images from existing images in our dataset. In the fourth stage, the transformed data are used to train models with the predetermined architectures. For this phase, the pretrained prediction models ResNet101, ResNext101, Se-ResNext50, ResNext50, and MobileNet V2 were used and compared. Finally, in the last stage, called “Interpretation/Evaluation”, metrics are used to evaluate the results obtained from each model, which are then analyzed and interpreted for subsequent decision-making.

### 3.2. Data Description

The dataset used was provided by the Peruvian Pterygium Center [7] and the authors of the articles [8,9,10,11]. A total of 540 anterior segment images of the eye (ASPIs) were obtained, of which 480 were captured using a medical tool called slit lamp and the rest using mobile devices. The data were classified into 3 classes, as shown in Figure 3: 101 ASPIs were classified as without pterygium, 167 as moderate pterygium, and 266 as severe pterygium. It should be noted that this data classification process was carried out by an ophthalmic surgeon from the Peruvian Pterygium Center.

### 3.3. Convolutional Neural Network Architecture

#### 3.3.1. ResNet101

ResNet, also known as Residual Network, was one of the first convolutional neural networks to use batch normalization. This architecture fine-tunes the network layers through residual functions directly linked to the inputs of each layer, facilitating efficient learning. Additionally, ResNet addresses a fairly common problem in deep networks called gradient disappearance, as it employs skip connections [21]. These skip connections allow gradients to bypass certain layers, thereby preventing the distortions that typically arise as the network depth and complexity increase. This approach enables the construction of deeper convolutional neural networks (CNNs) without compromising the accuracy.

The numerical suffix in the architecture’s name signifies the total number of layers; hence, ResNet101 comprises 101 layers. Out of these 101 layers, 4 convolutional blocks are allocated, and the layers are named the convolutional layer, max pooling layer, and average pooling layer, with exactly one layer each. These components are meticulously designed to uphold high precision in performance. The detailed structure of the ResNet101 architecture is illustrated in Figure 4. This sophisticated design allows ResNet101 to achieve remarkable accuracy in various computer vision tasks, making it the preferred choice for many deep learning applications.

#### 3.3.2. ResNext50, ResNext101, and Se-ResNext50

The ResNext architecture is characterized by its simple and highly modular structure. Introduced in 2016 as an enhancement of the ResNet50 architecture, ResNext incorporates the concept of “cardinality dimension”, which complements the traditional dimensions of depth and width in transformation strides [24]. This additional dimension enhances the architecture’s performance in specific tasks such as image classification by allowing the network to learn more diverse and discriminative features. In [6], it is mentioned that the structure consists of five distinct sections, including convolution blocks, which have three convolutional layers, and identity blocks, which consist of three transformation stages. This modular design allows for greater flexibility and efficiency in network training and performance. The architecture’s name includes a numerical suffix that indicates the total number of layers; for example, ResNext50 includes 50 layers, while ResNext101 includes 101 layers, providing increased depth and complexity. The detailed architecture of ResNext is depicted in Figure 5. This design ensures that the network can effectively handle complex tasks by leveraging its modular components and cardinality dimensions, making it a robust choice for various deep learning applications.

The Squeeze-and-Excitation (SE) block integrated into the ResNext50 architecture serves as a computational unit designed to transform inputs into feature maps, enhancing the compatibility with various convolutional neural network architectures and residual networks [21]. Despite introducing an additional computational overhead by being positioned before the addition operation, the SE block significantly improves the precision of the ResNext50 model when compared to the standard ResNet50 model. The SE block operates by recalibrating the network features through a two-step process: squeeze and excitation [21]. The “squeeze” step involves global average pooling to generate channel-wise statistics, while the “excitation” step leverages these statistics to recalibrate the feature responses via adaptive reweighting. This mechanism allows the network to emphasize the most important features dynamically, thereby enhancing the representational power and model performance.

Combining ResNext with SE blocks results in a more powerful architecture that excels in various computer vision tasks. The integration of SE blocks enables the network to adaptively highlight critical features, leading to improved model accuracy and robustness.

#### 3.3.3. MobileNet V2

MobileNet V2 is a neural network architecture optimized for mobile and embedded systems [18], designed to deliver high accuracy with minimal computational costs. It achieves efficiency by employing depthwise separable convolution, a method that reduces the number of parameters and operations needed compared to traditional convolutions. The architecture introduces an inverted residual structure, where the layers are organized to maintain performance while keeping the model lightweight. This structure, combined with bottleneck layers and average pooling, allows MobileNet V2 to be used effectively in transfer learning, particularly for vision tasks in resource-limited environments.

### 3.4. Valuation Metrics

Considering the established formulas in [25], commonly used in the implementation of artificial intelligence models, the evaluation of the results from the mentioned five models utilized metrics such as sensitivity, accuracy, precision, and the F1 score, which are detailed in Table 1.

The first metric mentioned is F1, which is the weighted harmonic mean of Pr and Rc, providing a balance between these two metrics [26]. The value of F1 is proportional to the predictability and reliability, i.e., a high value of F1 indicates the higher predictability and reliability of the system. Precision (Pr), the second metric, evaluates the direct proportion of true positive predictions (TP) among the total positive predictions made considering the true positives (TP) and false positives (FP). The third metric, called recall or sensitivity (Rc), allows one to correctly identify the current positive cases within the dataset. It considers true positives (TP) and false negatives (FN), with a high recall value indicating the model’s effectiveness in detecting most positive cases and minimizing false negatives. The final metric mentioned is the accuracy, which assesses the overall precision of the model in processing classification tasks. The accuracy is calculated by dividing the number of correct predictions by the total number of predictions, encompassing both true positives and true negatives (TN). This metric provides a straightforward measure of the model’s performance, reflecting its ability to make accurate predictions across all classes. Together, these metrics offer a comprehensive evaluation of the model’s performance, ensuring a thorough understanding of its strengths and areas for improvement in various classification tasks.

### 3.5. Dataset Prepration

A custom ASPI dataset was created to classify the severity level of pterygium if present. These ASPIs were pre-classified and labeled by an ophthalmologist to ensure the reliability of the results. From this classification, 70% of the total data were assigned to the training dataset, 15% to the test dataset, and 15% to the validation dataset. Figure 6 provides the detailed distribution of the dataset for each class across these three sets. It is important to note that all images were normalized to a resolution of 224 pixels by 224 pixels to ensure consistency in the model input.

To enhance the model’s generalization ability, a data augmentation approach was adopted, incorporating brightness and contrast adjustments, Gaussian noise, sharpening, gamma variation, and RGB channel manipulation, as seen in Figure 7. This technique mimics diverse capture conditions, replicating the variability found in images taken with a smartphone camera. By introducing such variability, the model learns more robust patterns, improving its generalization to new images. Standard methods like horizontal or vertical flips fail to capture the complexity of ophthalmic image variability, potentially limiting the prediction accuracy in real-world scenarios. Additionally, class weighting was employed to address class distribution imbalances, assigning specific weights during training to prevent bias towards prevalent pterygium severity categories.

## 4. Results

For this study, five pretrained ImageNet convolutional neural network (CNN) architectures were selected: ResNet101, ResNext101, ResNext50, Se-ResNext50, and MobileNet V2. Moreover, their weights were fine-tuned to leverage their pre-existing knowledge for the specific task of pterygium detection. The fine-tuning technique allowed the models to adapt to the new dataset while retaining the valuable features learned from the large-scale ImageNet dataset. The performance of these architectures was rigorously evaluated to determine the most effective model for implementation in the proposed mobile solution for pterygium detection in [6]. The training process used a categorical cross-entropy loss function, which is well suited for multi-class classification tasks, such as the three-class problem in this study. A batch size of 32 was used, along with 35 epochs to ensure adequate learning while preventing overfitting. The softmax function was utilized in the output layer to generate probability distributions for the classes, and the Adam optimizer was chosen for its efficiency and adaptive learning rate capabilities.

The proposed models were rigorously evaluated using several metrics, including the F1 score, precision, recall, and accuracy. The F1 score, which is a crucial measure of a model’s performance, varies across the different classes: without Pterygium, moderate pterygium, and severe pterygium. Specifically, for the class without pterygium, the Se-ResNext50 model achieved the highest F1 score of 97%. Similarly, for the moderate pterygium class, Se-ResNext50 again scored the highest, with an F1 score of 87%. For the severe pterygium class, Se-ResNext50 outperformed the other models, with an F1 score of 93%. Table 2 presents the F1 scores with data augmentation for each model, highlighting that Se-ResNext50 consistently achieved the highest average F1 score of 92% compared to the other models. This indicates that Se-ResNext50 is the most effective in classifying the severity levels of pterygium, benefiting significantly from data augmentation techniques, which enhance the model’s ability to generalize across varied and complex input data.

The precision metric, which measures the accuracy of the positive predictions made by the models, revealed notable results across different classes. For the class without pterygium, the Se-ResNext50 model achieved the highest precision value of 100%, indicating perfect accuracy in identifying instances without pterygium. In the moderate pterygium class, the ResNext50 model excelled, with the highest precision score of 90%, demonstrating its effectiveness in accurately predicting moderate cases. For the severe pterygium class, ResNet101 obtained the highest precision value of 94%, showcasing its capability in correctly identifying severe cases. Table 3 provides a detailed comparison of the precision scores for each model with data augmentation. The Se-ResNext50 model stands out with the highest average precision value of 93%, surpassing the other architectures.

The recall metric, which evaluates the model’s ability to correctly identify true positive cases, yielded significant insights across different classes. For the class without pterygium, the MobileNet V2 model achieved the highest recall value of 100%, demonstrating its effectiveness in correctly identifying instances without pterygium. In the moderate pterygium class, Se-ResNext50 excelled, achieving the highest recall score of 88%, indicating its capability in accurately detecting moderate cases. For the severe pterygium class, the ResNext50 model obtained the highest recall value of 98%, showcasing its proficiency in correctly identifying severe cases. Table 4 presents a comprehensive comparison of the recall scores for each model with data augmentation. The Se-ResNext50 model consistently achieved the highest average recall value of 92%, outperforming the other models. This high average recall value highlights the model’s ability to minimize false negatives and ensure that most positive cases are correctly identified across different classes.

The results showed that ResNet101, ResNext50, ResNext50, Se-ResNext50, and MobileNet V2 achieved accuracy of 84%, 88%, 76%, 92%, and 82%, respectively. Figure 8 illustrates the training performance based on the accuracy across all preset epochs. The accuracy trends, shown in Figure 8a–e, indicate the proportion of correct predictions. The constant increase and the absence of sudden drops to avoid overfitting or misfitting demonstrate the superiority of Se-ResNext50 over the other models.

All metrics already mentioned were calculated from the confusion matrix of each model, which is shown in Figure 9. These matrices have a tabular structure, where the rows represent the classes detected by the model and the columns represent the actual classes. The off-diagonal values represent the classification errors, while the values on the main diagonal indicate the number of predictions that the model made correctly for each class type (without pterygium, moderate pterygium, and severe pterygium).

From the confusion matrices, for ResNet101 (a), the model accurately predicts severe pterygium (37/41) but struggles more with moderate pterygium (21/26) and cases without pterygium (14/16). ResNext50 (b) shows better accuracy in predicting moderate pterygium (18/26) and severe pterygium (37/41), with some misclassification in the without pterygium category (11/16). ResNext101 (c) demonstrates balanced performance, with correct predictions for moderate pterygium (18/26), without pterygium (14/16), and severe pterygium (33/41), but shows a few more misclassifications than ResNext50. Se-ResNext50 (d) performs best in predicting moderate pterygium (22/26) and severe pterygium (40/41), with minimal misclassifications across all categories. Finally, MobileNet V2(e) most accurately predicts severe pterygium (33/41) and no pterygium (14/16), with some difficulty for moderate pterygium (18/26). Overall, ResNext50 and Se-ResNext50 show the highest accuracy and balanced performance across all pterygium severity levels.

## 5. Discussion

The results show that the model trained with Se-ResNext50, which obtained scores of 0.92, 0.93, 0.92, and 0.92 in terms of the F1 score, precision, recall, and accuracy, respectively, as detailed in Table 2, Table 3 and Table 4, was superior to the others. These results demonstrate that the Se-ResNext50 architecture outperformed the other evaluated architectures in the classification process of ophthalmological diseases, which is consistent with the results observed in [20,21,22,23].

These findings suggest that the inclusion of SE blocks in the ResNext50 architecture significantly enhances its ability to detect and classify pterygium, making Se-ResNext50 a robust candidate for implementation in the ophthalmological diagnostic mobile application proposed in [6], where ResNext50 was used.

This change in architecture represents a significant advancement in the accuracy and efficacy of pterygium diagnosis. Since Se-ResNext50 has demonstrated superior performance in terms of precision, recall, and accuracy, its implementation in the mobile application will improve the ability of healthcare professionals to detect and classify pterygium more reliably. This will not only optimize the early diagnosis of the condition but also reduce the incidence of false positives and negatives, which is crucial to avoid unnecessary treatments or, alternatively, the omission of essential interventions. Furthermore, by incorporating this advanced architecture in mobile applications, ophthalmologists and general practitioners could be equipped with a highly reliable tool that aids in real-time decision-making, reducing the reliance on costly and time-consuming manual assessments. This integration could lead to a shift toward more data-driven and AI-assisted diagnostic protocols in ophthalmology, allowing for earlier intervention and personalized treatment plans. Furthermore, the increased diagnostic accuracy of Se-ResNeXt50 has the potential to minimize diagnostic discrepancies and improve patient outcomes by ensuring that conditions are accurately identified early on. This could prevent the progression of pterygium to more severe stages that might require invasive procedures, thereby improving the overall quality of care.

Despite the promising results with Se-ResNext50, this study faces limitations related to the dataset, which was the same as that used in [6]. As noted in the previous research, the current dataset could be larger and more diverse to ensure the robust generalization of the model in diverse populations. The lack of a sufficiently large public dataset limits the model’s ability to adapt to different clinical scenarios. Additional data from hospitals and clinics need to be collected, and collaborations with researchers who have worked on pterygium detection need to be explored. Diversity in data, especially those captured with mobile devices, is essential to improve the reliability and applicability of the model in real-world situations.

To address the above limitations, future studies should focus on expanding and diversifying the dataset used in training. It is essential to collect a larger volume of data from multiple sources and different clinical contexts, which will allow for the better validation and generalization of the proposed model. Likewise, it is recommended to evaluate the performance of Se-ResNext50 in real clinical environments, which would not only allow for the verification of its effectiveness in practice, but also enable us to identify and overcome challenges related to the integration of the model into medical workflows and acceptance by healthcare professionals.

## 6. Conclusions

The benchmarking of CNN architectures demonstrates that Se-ResNext50 offers the best performance in pterygium detection, outperforming ResNet101, ResNext101, ResNext50, and MobileNet V2 in terms of precision, recall, and accuracy. These results not only support the implementation of Se-ResNext50 in the mobile application proposed in [6], but also underline its potential to significantly improve the diagnostic efficacy in ophthalmological medical centers. Adopting this architecture could significantly enhance the ability of healthcare professionals to make accurate and timely diagnoses, particularly in resource-limited settings. Furthermore, this advancement opens the door to exploring its application in other detectable visual diseases, extending its utility beyond pterygium. Future studies will focus on the further optimization of the model and the expansion of the dataset, which will allow for better generalization and a greater clinical impact. Its integration into medical workflows will also be evaluated, ensuring that the technology is not only accurate but also practical and accepted by healthcare professionals.

## Figures and Tables

**Figure 1 diagnostics-14-02026-f001:**
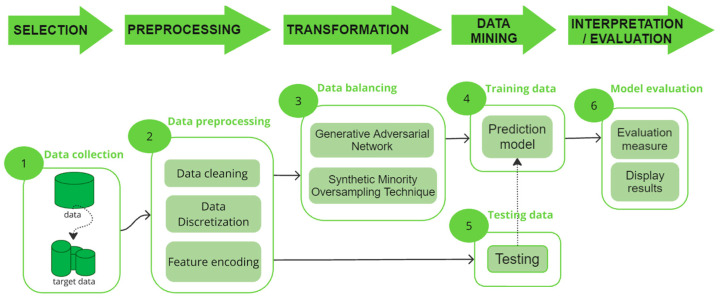
Stages of the KDD methodology.

**Figure 2 diagnostics-14-02026-f002:**
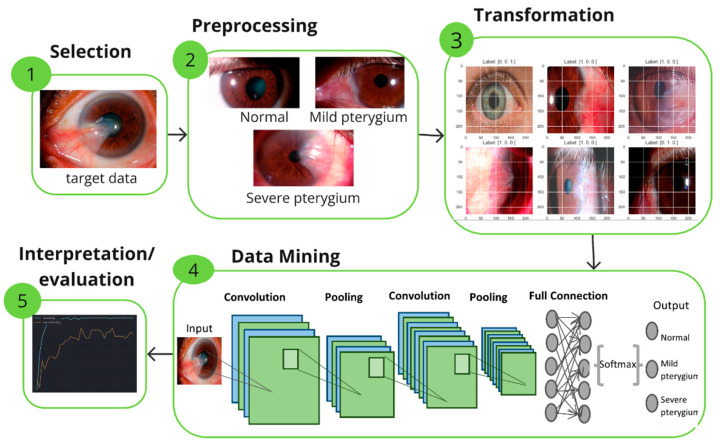
Stages of the KDD methodology used for model implementation.

**Figure 3 diagnostics-14-02026-f003:**
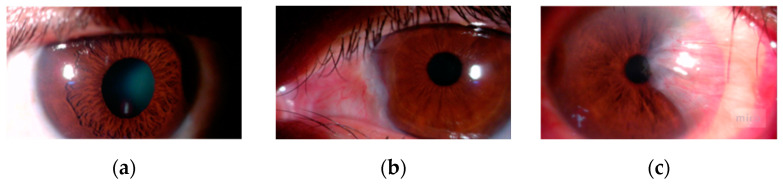
Image classification: (**a**) without pterygium, (**b**) moderate pterygium, (**c**) severe pterygium.

**Figure 4 diagnostics-14-02026-f004:**
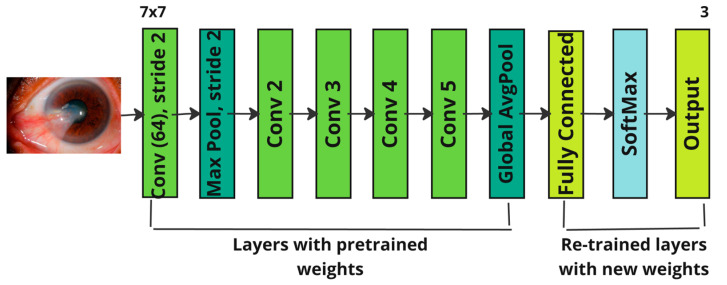
Architecture of ResNet101.

**Figure 5 diagnostics-14-02026-f005:**
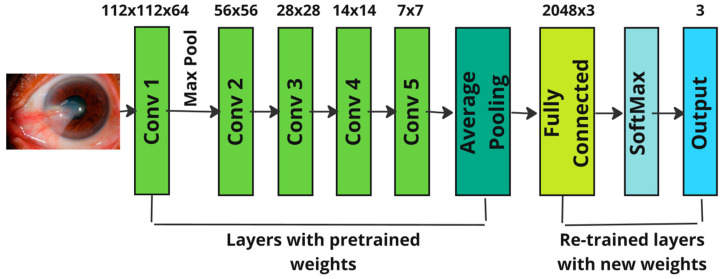
Architecture of ResNext.

**Figure 6 diagnostics-14-02026-f006:**
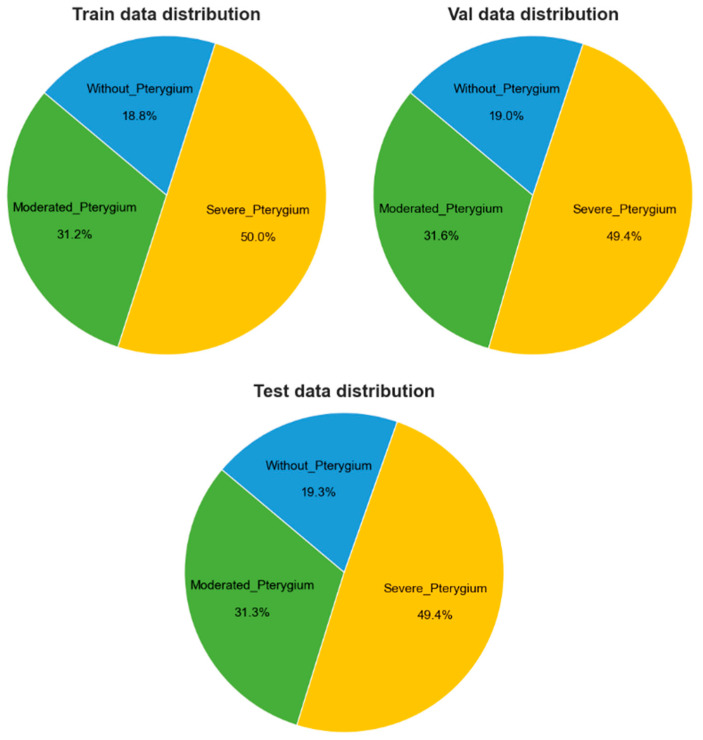
Data classification.

**Figure 7 diagnostics-14-02026-f007:**
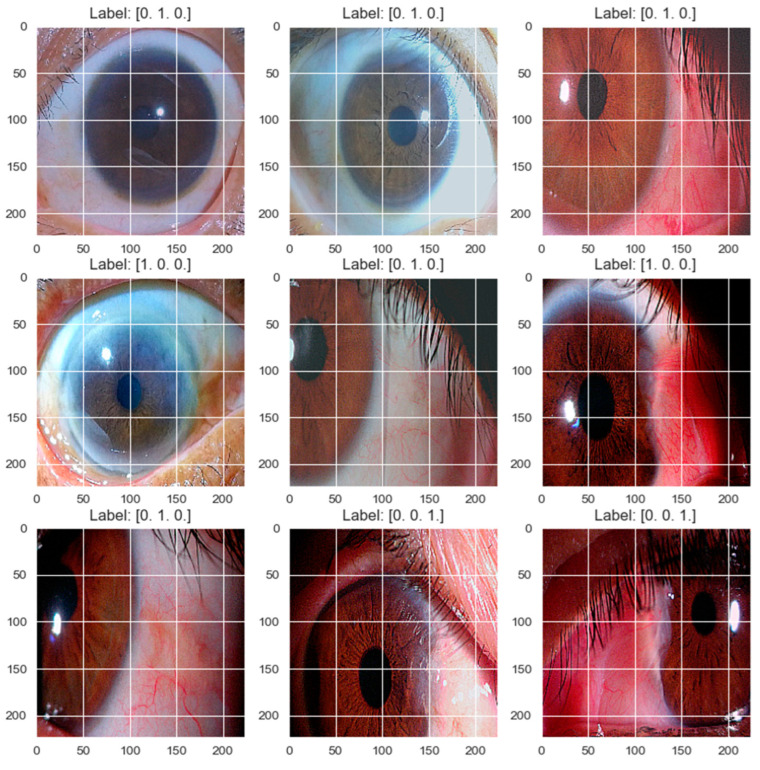
Variations in existing image data.

**Figure 8 diagnostics-14-02026-f008:**
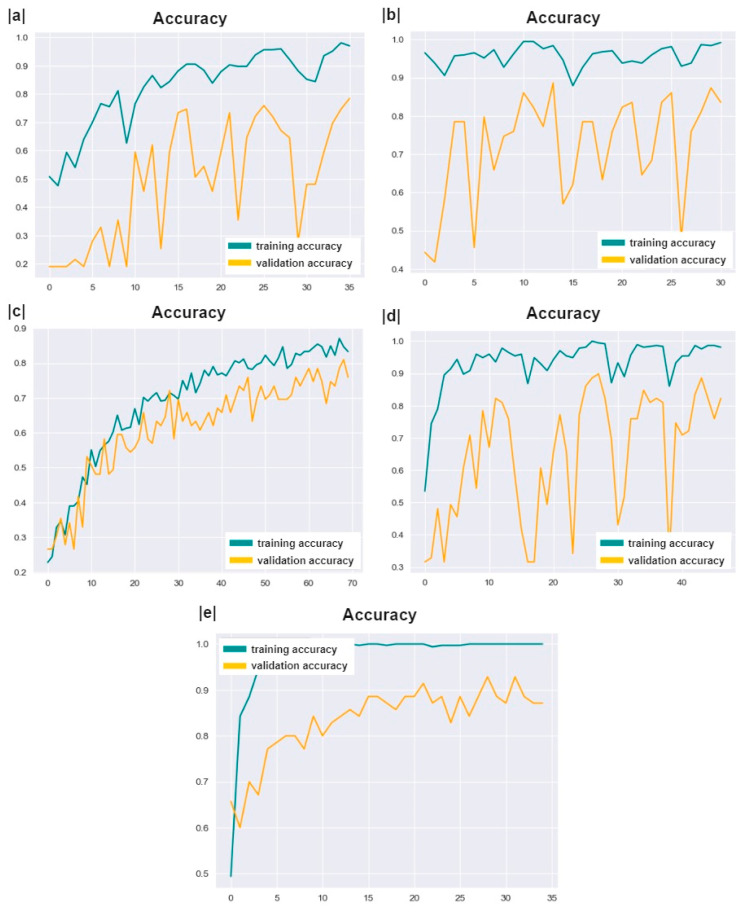
Accuracy (**a**–**e**) graphs for ResNet101, ResNext50, ResNext101, Se-ResNext50, and MobileNet V2 architectures, respectively.

**Figure 9 diagnostics-14-02026-f009:**
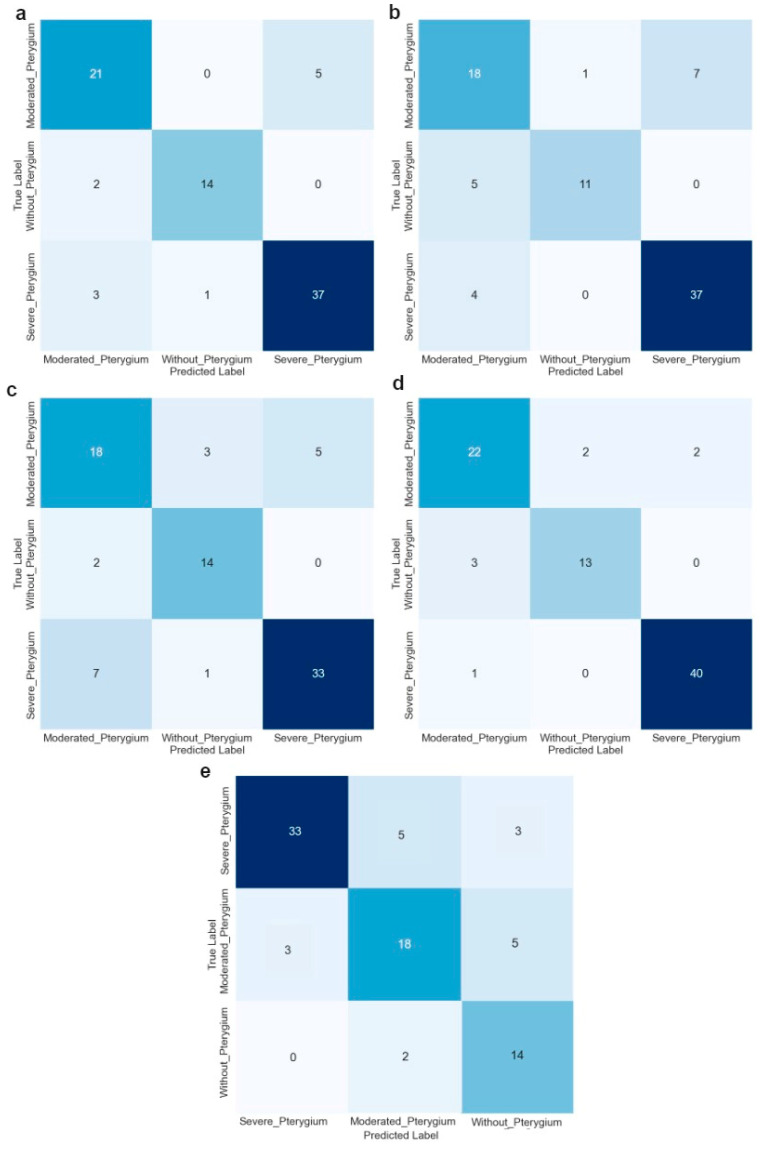
Confusion matrices for the architectures: (**a**) ResNet101, (**b**) ResNext50, (**c**) ResNext101, (**d**) Se-ResNext50, and (**e**) MobileNet V2.

**Table 1 diagnostics-14-02026-t001:** Metrics used for evaluation.

Metric	Formula
F1 Score (F1)	(2 × Pr × Rc) ÷ (Pr + Rc)
Precision (Pr)	TP ÷ (TP + FP)
Recall or Sensitivity (Rc)	TP ÷ (TP + FN)
Accuracy (Acc)	(TP + TN) ÷ (TP + TN + FP + FN)

**Table 2 diagnostics-14-02026-t002:** F1 scores of each architecture.

Architecture	Class			Total Score
	Without Pterygium	Moderate Pterygium	Severe Pterygium	
ResNet101	0.85	0.79	0.88	0.84
ResNext101	0.73	0.64	0.84	0.74
ResNext50	0.90	0.81	0.91	0.87
Se-ResNext50	0.97	0.87	0.93	0.92
MobileNet V2	0.86	0.74	0.86	0.82

**Table 3 diagnostics-14-02026-t003:** Precision of each architecture.

Architecture	Class			Total Score
	Without Pterygium	Moderate Pterygium	Severe Pterygium	
ResNet101	0.82	0.73	0.94	0.83
ResNext101	0.71	0.67	0.83	0.74
ResNext50	0.93	0.90	0.85	0.89
Se-ResNext50	1	0.85	0.93	0.93
MobileNet V2	0.75	0.80	0.87	0.81

**Table 4 diagnostics-14-02026-t004:** Recall of each architecture.

Architecture	Class			Total Score
	Without Pterygium	Moderate Pterygium	Severe Pterygium	
ResNet101	0.88	0.85	0.83	0.85
ResNext101	0.75	0.62	0.85	0.74
ResNext50	0.88	0.73	0.98	0.86
Se-ResNext50	0.94	0.88	0.93	0.92
MobileNet V2	1	0.70	0.85	0.85

## Data Availability

The data used in this study can be obtained from the corresponding author upon request.

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
