# Peer review of "A Performance Evaluation of Convolutional Neural Network Architectures for Pterygium Detection in Anterior Segment Eye Images"

_diagnostics, 2024, doi:10.3390/diagnostics14182026_

Round 1
Reviewer 1 Report
Comments and Suggestions for Authors
1- The study uses a dataset that, while relevant, may not be large or diverse enough to ensure the robustness of the findings across different populations and clinical scenarios. The limited dataset might affect the generalizability of the model, particularly in diverse real-world settings.
2- Include comparisons with other state-of-the-art models, such as more recent CNN architectures or different types of deep learning models, to provide a broader context for the results.
3- The following publications on deep learning and medical images may give you ideas for developing this publication
https://doi.org/10.1016/j.bspc.2019.101734
https://doi.org/10.18280/ts.400232
Author Response
|
Response to Reviewer 1 Comments
|
||
|
1. Summary |
|
|
|
Thank you very much for taking the time to review this manuscript. Please find the detailed responses below and the corresponding corrections highlighted in the re-submitted files.
|
||
|
2. Questions for General Evaluation |
Reviewer’s Evaluation |
Response and Revisions |
|
Does the introduction provide sufficient background and include all relevant references? |
Can be improved |
Adjustments have been made to the introduction (Page 2, third paragraph of the introduction, line 54-72) |
|
Is the research design appropriate? |
Can be improved |
Adjustments have been made to the research design |
|
Are the methods adequately described? |
Yes |
|
|
Are the results clearly presented? |
Yes |
|
|
Are the conclusions supported by the results? |
Can be improved |
Adjustments have been made to the conclusions |
|
3. Point-by-point response to Comments and Suggestions for Authors |
||
|
Comments 1: The study uses a dataset that, while relevant, may not be large or diverse enough to ensure the robustness of the findings across different populations and clinical scenarios. The limited dataset might affect the generalizability of the model, particularly in diverse real-world settings. |
||
|
Response 1: Thank you for pointing this out. We agree with this comment. However, in paragraph 4 and part of 5 of section '5 Discussion' we already talked about how difficult it is to find a sufficiently large set of public ASPIs and how important it is to have a larger and more diverse dataset for future research. Therefore, we have not made any changes. To find the correct section, please check: - Page 13, paragraph 4 and part of 5, lines 374-386
|
||
|
Comments 2: Include comparisons with other state-of-the-art models, such as more recent CNN architectures or different types of deep learning models, to provide a broader context for the results. |
||
|
Response 2: Agree. We have, accordingly, added a 5th model in the comparison: MobileNet V2. This was included in model definitions, results and conclusion. References related to this fifth model are found in: - Page 6, paragraph 3, lines 203-212 - Page 9, table 2 and 3 - Page 9, paragraph 4, line 302 - Page 10, table 4 - Page 10, paragraph 5, line 314-315 - Page 11, figure 8, line 323 - Page 12, paragraph 6, lines 338-340 - Page 13, figure 9, line 345 - Page 13, paragraph 1, line 394 |
||
Comments 3: The following publications on deep learning and medical images may give you ideas for developing this publication: https://doi.org/10.1016/j.bspc.2019.101734, https://doi.org/10.18280/ts.400232
Response 3: Agree. Suggestion has been considered.

Reviewer 2 Report
Comments and Suggestions for Authors
Comments and Suggestions for Authors:
This manuscript presents a thorough and well-structured study on the evaluation of various Convolutional Neural Network (CNN) architectures for pterygium detection in anterior segment eye images. The research is highly relevant and contributes valuable insights to the field of ophthalmology, particularly in the application of deep learning for medical diagnostics.
Strengths:
- The introduction provides a comprehensive background on pterygium and the limitations of existing diagnostic methods, making a strong case for the use of AI in this domain.
- The research design is robust, with a clear explanation of the methodologies used, including the Knowledge Discovery in Databases (KDD) process and the evaluation metrics.
- The results are clearly presented, with detailed comparisons of the performance of different CNN architectures. The use of tables and figures enhances the clarity and understanding of the findings.
- The conclusions are well-supported by the results, particularly the superior performance of the Se-ResNext50 architecture, which is convincingly demonstrated through various metrics.
Suggestions for Improvement:
- While the introduction is solid, it could benefit from a more explicit identification of the research gap that this study addresses, further highlighting what differentiates this work from previous studies.
- The manuscript would benefit from minor editing to improve the flow and readability of the text. While the English language quality is generally good, there are some areas where sentence structure could be refined for clarity.
For example:
Here are a few areas where sentence structure could be refined for clarity:
-
Introduction Section:
- Original: "Although there are effective conventional methods for detecting pterygium, they have limitations in terms of time and cost, as they require specialized medical tools or equipment and an ophthalmologist with prior experience in the area to manually classify the affected areas in the eye images."
- Suggested Revision: "While conventional methods for detecting pterygium are effective, they are time-consuming and costly, requiring specialized medical tools and an experienced ophthalmologist to manually classify affected areas in eye images."
-
Methods Section:
- Original: "Knowledge Discovery in Databases (KDD) is a common term in the field of data mining and artificial intelligence. This methodology, illustrated in Figure 1, allows for the identification of useful patterns and knowledge from large volumes of data."
- Suggested Revision: "Knowledge Discovery in Databases (KDD), commonly used in data mining and artificial intelligence, is a methodology that identifies useful patterns and insights from large datasets, as illustrated in Figure 1."
-
Results Section:
- Original: "The training process employed a categorical cross-entropy loss function, which is ideal for multi-class classification tasks as was the case here with 3 classes."
- Suggested Revision: "The training process used a categorical cross-entropy loss function, which is well-suited for multi-class classification tasks, such as the three-class problem in this study."
-
Discussion Section:
- Original: "However, despite the promising results obtained with Se-ResNext50, the study still faces limitations related to the dataset used, which is the same one used in [6]."
- Suggested Revision: "Despite the promising results with Se-ResNext50, the study faces limitations related to the dataset, which was the same as that used in [6]."
-
Conclusion Section:
- Original: "By adopting this architecture, the ability of healthcare professionals to make more accurate and timely diagnoses is enhanced, especially in resource-limited settings."
- Suggested Revision: "Adopting this architecture could significantly enhance the ability of healthcare professionals to make accurate and timely diagnoses, particularly in resource-limited settings."
-
- In the discussion section, consider elaborating more on the potential clinical implications of the findings, particularly how the integration of the Se-ResNext50 architecture could change current practices in ophthalmology.
- Given that the dataset size is a noted limitation, future studies should focus on expanding the dataset and validating the models in more diverse clinical settings. This would enhance the generalizability of the findings.
Conclusion: This is a strong and valuable contribution to the field. With minor revisions and some additional clarity in certain sections, this manuscript will be a solid candidate for publication.
Comments on the Quality of English LanguageThe overall quality of the English language in the manuscript is good, with clear and concise expression of ideas. However, there are a few areas where sentence structure and phrasing could be refined to enhance clarity and readability.
Minor revisions are suggested to improve the flow of the text and ensure that the content is easily understandable to a broad audience.
These adjustments will help to better convey the significance and findings of the research.
Author Response
|
Comments 1: While the introduction is solid, it could benefit from a more explicit identification of the research gap that this study addresses, further highlighting what differentiates this work from previous studies.
|
|
Response 1: Thank you for your comment. We agree with it, therefore, we do the following: The paragraph that was initially “In [6], a system for the detection of pterygium is proposed using a ResNext50 model. Therefore, this article presents the analysis and comparison of this model with others such as ResNet101, ResNext101, and Se-ResNext50. Additionally, the same dataset obtained from the Peruvian Pterygium Center [7] and external authors [8], [9], [10], [11] will be used. Of the four models evaluated with our data set (ResNet101, ResNext101, Se-ResNext50 and ResNext50), the Se-ResNext50 architecture achieved percentages of up to 97%, 100%, 94% and 92% in F1 score, precision, recall and accuracy, respectively” we updated it with “Despite the advancements achieved in pterygium detection using AI models, there are still limitations that have not been fully addressed. In particular, previous studies, such as the one presented in [6], focused on the development of a mobile application for pterygium detection using the ResNeXt50 architecture. However, this work did not consider a direct comparison with more recent models that hold the potential for superior performance but rather limited its evaluation to the architectures most commonly used by other authors. This gap is critical, as the effectiveness and accuracy of mobile applications in clinical settings directly depend on the robustness of the underlying AI model. In this context, our study is distinguished by evaluating and comparing the performance of more advanced CNN architectures, such as ResNet101, ResNeXt101, and Se-ResNeXt50, which have demonstrated superior results in other computer vision applications. This approach will allow us to identify which of these architectures offers the best performance and, therefore, has the greatest potential for integration into the previously developed mobile application, thereby enhancing its ability to detect pterygium efficiently, accurately, and in real-time. Using the same dataset obtained from the Peruvian Pterygium Center [7] and external authors [8], [9], [10], [11], among the five evaluated models (ResNet101, ResNeXt101, Se-ResNeXt50, ResNeXt50 and MobileNet V2), the Se-ResNeXt50 architecture achieved performance metrics of up to 97%, 100%, 94%, and 92% in F1 score, precision, recall, and accuracy, respectively”. The correction can be viewed in: - Page 2, third paragraph of the introduction, line 54-72
|
|
Comments 2: The manuscript would benefit from minor editing to improve the flow and readability of the text. While the English language quality is generally good, there are some areas where sentence structure could be refined for clarity. Here are a few areas where sentence structure could be refined for clarity: 1. Introduction Section: § Original: "Although there are effective conventional methods for detecting pterygium, they have limitations in terms of time and cost, as they require specialized medical tools or equipment and an ophthalmologist with prior experience in the area to manually classify the affected areas in the eye images." § Suggested Revision: "While conventional methods for detecting pterygium are effective, they are time-consuming and costly, requiring specialized medical tools and an experienced ophthalmologist to manually classify affected areas in eye images." 2. Methods Section: § Original: "Knowledge Discovery in Databases (KDD) is a common term in the field of data mining and artificial intelligence. This methodology, illustrated in Figure 1, allows for the identification of useful patterns and knowledge from large volumes of data." § Suggested Revision: "Knowledge Discovery in Databases (KDD), commonly used in data mining and artificial intelligence, is a methodology that identifies useful patterns and insights from large datasets, as illustrated in Figure 1." 3. Results Section: § Original: "The training process employed a categorical cross-entropy loss function, which is ideal for multi-class classification tasks as was the case here with 3 classes." § Suggested Revision: "The training process used a categorical cross-entropy loss function, which is well-suited for multi-class classification tasks, such as the three-class problem in this study." 4. Discussion Section: § Original: "However, despite the promising results obtained with Se-ResNext50, the study still faces limitations related to the dataset used, which is the same one used in [6]." § Suggested Revision: "Despite the promising results with Se-ResNext50, the study faces limitations related to the dataset, which was the same as that used in [6]." 5. Conclusion Section: § Original: "By adopting this architecture, the ability of healthcare professionals to make more accurate and timely diagnoses is enhanced, especially in resource-limited settings." § Suggested Revision: "Adopting this architecture could significantly enhance the ability of healthcare professionals to make accurate and timely diagnoses, particularly in resource-limited settings." Response 2: We agree with the comment and update the paragraphs with the suggestions made. The correction can be viewed in: - Page 1, first paragraph of the introduction, line 33-36 - Page 3, first paragraph of the Materials and Methods, line 121-123 - Page 8, first paragraph of the Results, line 268-270 - Page 13, fourth paragraph of the Discussion, line 374-375 - Page 14, first paragraph of the Conclusions, line 397-398
Comments 3: In the discussion section, consider elaborating more on the potential clinical implications of the findings, particularly how the integration of the Se-ResNext50 architecture could change current practices in ophthalmology.
Response 3: Thank you for your comment. We agree with it, therefore, we do the following: The paragraph that was initially “This change in architecture represents a significant advance in the accuracy and efficacy of pterygium diagnosis. Since Se-ResNext50 has demonstrated superior performance in terms of precision, recall and accuracy, its implementation in the mobile application will improve the ability of healthcare professionals to detect and classify pterygium more reliably. This will not only optimize the early diagnosis of the condition, but also reduce the incidence of false positives and negatives, which is crucial to avoid unnecessary treatments or, alternatively, the omission of essential interventions.” We updated it with “This change in architecture represents a significant advance in the accuracy and efficacy of pterygium diagnosis. Since Se-ResNext50 has demonstrated superior performance in terms of precision, recall and accuracy, its implementation in the mobile application will improve the ability of healthcare professionals to detect and classify pterygium more reliably. This will not only optimize the early diagnosis of the condition, but also reduce the incidence of false positives and negatives, which is crucial to avoid unnecessary treatments or, alternatively, the omission of essential interventions. Furthermore, by incorporating this advanced architecture in mobile applications, ophthalmologists and general practitioners could be equipped with a highly reliable tool that aids in real-time decision making, reducing the reliance on costly and time-consuming manual assessments. This integration could lead to a shift toward more data-driven and AI-assisted di-agnostic protocols in ophthalmology, allowing for earlier intervention and personalized treatment plans. Furthermore, the increased diagnostic accuracy of Se-ResNeXt50 has the potential to minimize diagnostic discrepancies and improve patient outcomes by ensuring conditions are accurately identified early on. This could prevent the progression of pterygium to more severe stages that might require invasive procedures, thereby improving the overall quality of care” - Page 13, third paragraph of the Discussion, line 357-373
Comments 4: Given that the dataset size is a noted limitation, future studies should focus on expanding the dataset and validating the models in more diverse clinical settings. This would enhance the generalizability of the findings
|
|
Response 4: We agree with the comment, no additional changes were made. However, in paragraph 4 and part of 5 of section '5 Discussion' we already talked about how difficult it is to find a sufficiently large set of public ASPIs and how important it is to have a larger and more diverse dataset for future research.
|
|
4. Response to Comments on the Quality of English Language |
|
Point 1: The overall quality of the English language in the manuscript is good, with clear and concise expression of ideas. However, there are a few areas where sentence structure and phrasing could be refined to enhance clarity and readability. |
|
Response 1: We appreciate your comment, we made the corresponding review and updated as suggested in comment 2 |
